# The Role of Demoralization and Hopelessness in Suicide Risk in Schizophrenia: A Review of the Literature

**DOI:** 10.3390/medicina55050200

**Published:** 2019-05-23

**Authors:** Isabella Berardelli, Salvatore Sarubbi, Elena Rogante, Michael Hawkins, Gabriele Cocco, Denise Erbuto, David Lester, Maurizio Pompili

**Affiliations:** 1Department of Neurosciences, Mental Health and Sensory Organs, Suicide Prevention Center, Sant’Andrea Hospital, Sapienza University of Rome, 00185 Rome, Italy; denise.erbuto@uniroma1.it (D.E.); maurizio.pompili@uniroma1.it (M.P.); 2Department of Psychology, Sapienza University of Rome, 00185 Rome, Italy; salvatore.sarubbi@uniroma1.it (S.S.); elena.rogante@gmail.com (E.R.); 3Department of Psychiatry, University of Toronto, Toronto, ON M4B 1B4, Canada; michael.hawkins@mail.utoronto.ca; 4Faculty of Medicine and Psychology, Sapienza University of Rome, 00185 Rome, Italy; gabri.cocco84@hotmail.it; 5Psychology Program, Stockton University, Galloway, NJ 08205, USA; David.Lester@stockton.edu

**Keywords:** demoralization, schizophrenia, suicide risk

## Abstract

*Background and Objectives:* Demoralization has been defined by hopelessness and helplessness attributable to a loss of purpose and meaning in life. Demoralization is a meaningful mental health concern, frequently associated with suicide risk in medical and psychiatric patients. The aim of this systematic review was to synthesize the recent empirical evidence on demoralization in patients with schizophrenia and to better understand the relationship between demoralization and suicide risk in patients with schizophrenia. *Methods:* A comprehensive literature search using key words and subject headings was performed following PRISMA guidelines with several bibliographic databases, resulting in the identification of 27 studies. *Results:* The findings suggested that demoralization is prevalent in patients with schizophrenia and supported the hypothesis that the association between depression and suicide is moderated by hopelessness. In clinical practice, it is important to recognize symptoms of demoralization using appropriate psychological tools to better understand the suffering of patients with schizophrenia and to implement suicide prevention programs.

## 1. Introduction

Jerome Frank [1], more than forty years ago, defined demoralization as “a syndrome of existential distress occurring in patients with severe conditions that threaten life or integrity of being, such as physical illness or mental disorders.” Demoralization is characterized by ‘‘feelings of impotence, isolation, and despair.” Frank interpreted demoralization as an outcome caused by someone’s failure to cope when faced with an event generally described as easy to manage [2]. In those who are demoralized, self-esteem suffers, and rejection is felt as a result of individuals believing they have failed to meet the expectations of others [1,2,3]. This highlights the salience that stressful events may have on the human psyche [4].

In more recent years, Irvin Yalom, an existential psychotherapist [5], noted how demoralization represents the outcome of a conflict between the individual and his personal existence. When a person is unable to face this conflict, demoralization can develop. Based on this, the definition of demoralization was subsequently enriched with new concepts [6], with the link between distress and subjective incompetence at the base of the construct of demoralization.

In the following years, several authors attempted to describe the demoralization syndrome. Fava et al. [7] used three diagnostic criteria for demoralization, including the failure of individuals to meet expectations set by themselves, those set by others, and their general inability to cope with demands. This results in feelings of helplessness, hopelessness, and a desire to give up. According to a psychosomatic perspective, demoralization is a predictor for the development of a medical illness [7].

A recent definition of demoralization was proposed by Kissane and Clarke [8,9] at the beginning of the 2000s. Demoralization was defined as a clinical entity characterized by symptoms such as existential distress, hopelessness, loss of meaning and purpose in life, a sense of being trapped, personal failure, and difficulty coping. Kissane focused on demoralization as a predictive variable for suicidal ideation in terminally ill patients [2]. Kissane described demoralization as an abnormal response characterized by two key aspects: loss of meaning in life and loss of hope. Considering that hopelessness is one of the most important predictors of suicide, patients with the demoralization syndrome may have a higher risk of suicide than patients without the demoralization syndrome [2].

The debate on demoralization as a syndrome is still controversial. The categorization of demoralization as an “abnormal response” has been criticized by different authors. Several authors emphasized that demoralization is a normal psychological response in patients affected by a medical illness [10] and compared demoralization to the Adjustment Disorders, described in DSM-5, or as a normal response to loss [11,12]. Furthermore, the main controversy centers on whether demoralization constitutes a syndrome of despair, distress, and hopelessness separate from depression. Kissane saw the concept of demoralization as a syndrome, contextualizing it on a spectrum where the end of this spectrum warrants a “psychiatric alert”. Demoralization becomes clinically relevant when there is a significant loss of meaning and purpose in life, and suicidal thinking appears [13]. It has been argued that the critical feature distinguishing depression from demoralization is the presence or absence of anhedonia [14]. However, depression and anxiety are symptoms that are often present in the demoralization syndrome [8].

Although demoralization can occur in response to many distressing circumstances, much of the literature about demoralization focuses on people who have a terminal or severely disabling physical illness [15,16], but only a few studies have focused on psychiatric patients, such as patients with schizophrenia [17].

Patients with schizophrenia often die prematurely [18]. Though the lifetime risk for suicide for people with schizophrenia is estimated to be about 4.9% [19], up to 40% [20] of their premature mortality can be attributed to suicide and unnatural deaths. Several psychological features seem to contribute to suicide in this population, including hopelessness, recent loss or rejection, fear of further illness deterioration, awareness of their illness, loss of faith in treatment, post-psychotic depression, or failure in interpersonal relationships [21]. A number of clinical features have been associated with completed suicide in people with schizophrenia, including being young, male, Caucasian, and never married, and having good premorbid function and a history of substance abuse and suicide attempts [21,22]. Hopelessness, social isolation, and hospitalization are also important risk factors for completed suicide in individuals with schizophrenia. Among these features, hopelessness plays a large role in affecting the suicide risk of these patients. Higher hopelessness scores, regardless of the presence of only depressive symptoms, predicted a worse short-term outcome and worse global functioning, which are considered important risks factor for suicidal behavior [23]. Furthermore, a greater surveillance for suicide is required in the period after discharge from hospital and after periods of remission. Patients with schizophrenia usually experience hopelessness and demoralization after discharge from hospital which increases their suicide risk. During periods of remission, they can develop a painful awareness of their illness. At this point, their expectations contrast sharply with their declining functioning, leading to feelings of inadequacy, depression, and hopelessness. It has been suggested that hopelessness and greater insight are associated with current and lifetime risk of suicide [24].

Drake has described a demoralization syndrome in individuals with schizophrenia, in which repeated exacerbations of psychotic symptoms, functional deterioration, and a non-delusional awareness of the effects of the illness can lead to feelings of hopelessness, depression, and ultimately suicide [25].

In this view, premorbid adjustment and insight interact, resulting in demoralization and depression, potentially culminating in suicidal behavior. Interestingly, on the one hand, depression has been recognized as a major risk factor for suicide attempt among patients with schizophrenia [26]. On the other hand, depression is difficult to diagnose in patients with schizophrenia. In clinical practice, nonspecific sadness is more often observed than major depressive episodes. In the 1980s, Bartels and Drake [27] considered chronic demoralization to be a variable that predicted the risk of suicide in patients with schizophrenia. This hypothesis was then validated in several studies that have shown how the presence of depression, hopelessness, negative self-thoughts, anxiety, insomnia, self-devaluation, low self-esteem, and a feeling of guilt were strongly associated with a higher risk of suicide in patients with schizophrenia [21,22,23,24,25,26,27,28]. Research on this topic was then enriched with papers on patients with schizophrenia linking hopelessness with social and vocational dysfunction [29,30,31], suicide risk [25,26,27,28,29,30,31,32], avoidant coping [33,34], and stigma [35,36,37].

In line with the hypothesis that the demoralization syndrome and hopelessness are important risk factors for suicide in patients with schizophrenia, and on the basis that these two constructs are related to both patient insight and stigma, variables involved in suicide risk, we reviewed those studies investigating the presence of demoralization (and its constructs) and suicide, in patients with schizophrenia.

The principal aim of this review was to clarify the role of demoralization and hopelessness in affecting suicide risk in patients with schizophrenia. The second aim of this review paper was to better understand the complex relationship between demoralization and hopelessness, along with patient insight and stigma, and suicide risk in patients with schizophrenia.

## 2. Methods

We performed a systematic review of demoralization (according to a definition of demoralization that includes the inability to cope, helplessness, hopelessness, and low self-esteem) and suicidal risk in schizophrenia using MedLine, Excerpta Medica, PsycLit, PsycInfo, and Index Medicus search to identify all papers published between 1970 and 2018. The PRISMA statement for reporting systematic reviews was followed. Search terms used were: demoralization OR demoralization syndrome OR helplessness OR hopelessness AND suicide risk AND Schizophrenia, Psychosis. We first reviewed the titles and abstracts and applied the selection criteria outlined above with the exception of study design (Figure 1). Only articles published in English peer-reviewed journals were considered, and the articles were examined to for their relevance based on the inclusion criteria. Possible discrepancies between the reviewers with regard to inclusion criteria were resolved through consultation with the senior authors. In addition, reference lists were also examined. In the results section, we excluded case reports, meta-analyses, and systematic reviews, and studies that did not clearly report statistical analysis, diagnostic criteria, or the number of patients included. We found 27 articles on demoralization and schizophrenia that met these criteria and we discussed them in the two sections of the results (Clinical Studies included in qualitative synthesis). We have included three systematic reviews exclusively in the discussion section but not in the results section. The principal reviewer (IB) checked all items. Then, three reviewers independently inspected all citations of the studies identified by the search and grouped them according to topic.

## 3. Results

### 3.1. Demoralization and Suicide Risk in Schizophrenia

Depressive features, including demoralization and hopelessness, are common in schizophrenia and are often intertwined with psychotic symptoms, becoming a significant mediator of disability and suicidality. Cotton et al. [38] interviewed 20 therapists who had clinically followed 20 patients with schizophrenia who completed suicide. These interviews indicated that the profile of the patients was of young men in their thirties with a chronic history of illness with exacerbations. From the interviews, it emerged that patients expressed hopelessness at the time of their suicide. The patients had a strong desire to escape through death, and most of them had a previous history of suicidal behavior. Cotton then showed how different clinical features resulted in guidelines for the treatment of these patients. For example, it is important to assess the patient’s self-esteem, the protective function that psychosis may have in regard to suicide, and the importance of differentiating an inability to function from an unwillingness to function (i.e., unwillingness to share their thoughts of life and death or share the burden of despair resulting from their illness).

Drake et al. [39] examined risk factors for suicide in patients with schizophrenia, noting that the risk factors for suicide included demoralization and feelings of hopelessness and inadequacy. A few years later, the same authors investigated the clinical features of depression, hopelessness, and their relationship to major depressive episodes and suicides in 104 patients with schizophrenia, 15 of whom died by suicide [40]. Most patients (54%) endorsed “depressed mood”, and 21% of these patients met DSM criteria for a depressive disorder. Although the presence of depressed mood increased their risk of suicide, the severity of depression did not increase the risk of suicide. In contrast, the development of hopelessness, in addition to depressed mood, significantly increased the probability of suicide. In absence of hopelessness, depressed patients with schizophrenia presented a similar risk for suicide to that of non-depressed patients with schizophrenia. Thus, these results suggest that hopelessness mediates the relationship between depression and suicide risk in patients with schizophrenia.

Data from a prospective community treatment study on suicide in patients with schizophrenia showed that the group of patients who died by suicide were younger, with an earlier onset of illness compared to patients who did not complete suicide, and suicide occurred very early in the course of illness [32]. Furthermore, hopelessness was a very discriminative feature between the two groups (z = 3.27, *p* < 0.05), and patients who completed suicide showed a significant increase in hopelessness and depression.

Fenton et al. [41] studied the relationship between positive and negative symptoms, illness subtype, and suicidal behavior in 187 patients with schizophrenia, 87 patients with schizoaffective disorder, 15 patients with schizophreniform disorder, and 33 patients with schizotypal personality disorder. The authors demonstrated that the progressive loss of social drive, the diminished capacity to experience affect, and an indifference toward the future may preclude the painful self-awareness associated with suicide.

Nordentoft et al. [42] investigated predictive factors for suicidal behavior in 321 patients with first-episode psychosis. Their results demonstrated that a suicide attempt was associated with younger age, depression, hopelessness, and hallucinations. Being female, reported hopelessness at baseline, the presence of hallucinations, and a suicide attempt reported at the initial interview were associated with a suicide attempt by the one-year follow up.

Kim et al. [43] enrolled 333 patients with schizophrenia to clarify the clinical role of hopelessness, insight, and cognitive dysfunction for suicide risk. They found that hopelessness was significantly higher in patients with lifetime suicidality and in those with current suicide risk compared to patients without history of suicidal behavior, thereby supporting the role of hopelessness as a predictor of the lifetime risk of suicide.

Montross et al. [44] examined the prevalence of suicidal behavior in 132 patients with schizophrenia. Depression and hopelessness were significantly higher in the group with suicidal ideation. Predictive features for current suicidal ideation included depression, hopelessness, gender, general psychopathology, and the presence of a past suicide attempt. They also found that hopelessness and a past history of suicide attempts were the only variables associated with current suicidal ideation.

Restifo et al. [45] tested the demoralization hypothesis as an independent risk factor for suicidal behaviors in 164 patients, 115 of whom had a diagnosis of schizophrenia and 49 a diagnosis of schizoaffective disorder. The sample was divided in two groups: patients with a history of a previous suicide attempt (N = 59) and patients without a history of previous suicide attempts (N = 105). The authors observed that premorbid functioning and insight were associated with more depressive symptoms. Furthermore, cognitive symptoms of depression (poor concentration, indecisiveness, forgetfulness) and subclinical symptoms of depression were linked with past suicide attempts, while hopelessness predicted current and lifetime risk of suicide among these patients with schizophrenia. In support of these results, a case-control study [46] confirmed that a history of suicide attempts, hopelessness, and self-devaluation were the three variables that had the strongest association with completed suicide in patients with schizophrenia.

More recently Klonsky, et al. [47], in a 10-year longitudinal study of 414 patients with first-admission psychosis, clarified the relationship between hopelessness and suicide attempt. Both hopelessness and recent suicide attempts were assessed at multiple time-points. The results demonstrated that hopelessness at baseline predicted a suicide attempt during the subsequent 10 years independent of the presence of depression.

A few studies have investigated the association between self-esteem and suicide risk in patients with schizophrenia [48] (Table 1). Tarrier et al. [49] examined 59 patients with recent onset schizophrenia, reporting that greater hopelessness (odds ratio (OR) 1.22) and a longer duration of illness (OR 1.13) increased the risk of suicide in these patients. Hopelessness was associated with higher negative self-evaluation and social isolation, while negative self-evaluation was associated with self-criticism and negative symptoms. Acosta et al. [50], in a study of 60 patients with schizophrenia showed that negative cognitions about the psychiatric diagnosis were associated with depressive symptoms and hopelessness. Interestingly, negative cognition did not differ in patients with and without a history of previous suicide attempts. Yoo et al. [51] studied 87 patients with schizophrenia, of whom 20 (23.0%) had a history of suicide attempts. Their results revealed that patients with a history of suicide attempts had significantly higher scores on the Beck Depression Inventory (*p* = 0.036) and on the Korean version of the Internalized Stigma of Mental Illness scale (*p* = 0.009), and significantly lower scores on the Rosenberg Self-Esteem Scale (*p* = 0.001), demonstrating that low self-esteem represents a psychological feature of those who attempt suicide, thereby increasing their risk of suicide.

Five hundred and ten individuals with schizophrenia were studied by Ran et al. [52] with the aim of identifying clinical variables that could discern between patients with a history of suicide attempts and those without a history of suicide attempts. Of the 510 patients, 60.5% of patients that attempted suicide endorsed depressive symptoms and hopelessness. The suicide attempters also had more auditory hallucinations, delusions, and positive symptoms and presence of lifetime hopelessness.

### 3.2. Demoralization, Insight, and Stigma in Schizophrenia

How depression, demoralization, hopelessness, positive and negative symptoms, illness subtype, and suicidal tendency relate to one another in patients with schizophrenia is poorly understood but of great clinical relevance. Several studies have highlighted the link between high levels of insight, low self-esteem, and the quality of life [53,54] (Table 2). Patients with psychiatric illnesses experience discrimination, less life satisfaction, stigma, and feel demoralized and rejected by others [55]. Recent research has suggested that self-stigma results in reduced self-esteem, depression, and anxiety and hinders recovery [56,57]. In a cross-sectional study on 85 patients with schizophrenia receiving maintenance therapy, Birchwood et al. [58] investigated the relationship between depression, the acceptance or rejection of mental illness, the perceived controllability of illness and the acceptance of cultural stereotypes. Twenty-nine per cent of the patients with schizophrenia were considered ‘depressed’ using the cut-off point of 15 on the Beck Depression Inventory (BDI).

The relationship between hopelessness and global functioning was investigated by Aguilar et al. [59] in first-episode psychosis patients. They found that patients with first-episode schizophrenia had higher levels of hopelessness than non-schizophrenia patients, and that higher hopelessness scores predicted worse global functioning at a one-year follow-up in patients with psychosis. In a cross-sectional study, Carroll et al. [60] explored the relationship between hopelessness and level of insight in 28 forensic patients with schizophrenia, testing the hypothesis that insight domains of ‘compliance with treatment’ and ‘awareness of illness’, as evaluated by the Schedule for Assessment of Insight (SAI), would be positively correlated with hopelessness. They showed that awareness of illness, but not compliance with the treatment, was positively correlated with level of hopelessness (i.e., patients who had a poor awareness of their illness generally had more hope for their future).

Lysaker et al. [61] investigated whether expectations about the future and motivation to persist (two aspects of hope) were correlated with neurocognitive patterns, personality, symptoms, and social functioning among 52 patients in a post-acute phase of schizophrenia. The authors suggested that both these aspects of hope were linked to lesser levels of stigma, fewer symptoms of schizophrenia, lesser anxiety, and lesser preference for avoidant forms of coping, emphasizing how different aspects of stigma are possibly linked to different domains of hope.

White et al. [62], in a study on 100 patients with schizophrenia, investigated the psychological features of non-hopeless patients (including personal beliefs about illness) and versus hopeless patients. Of the total sample, 25% of patients reported severe hopelessness, and hopelessness correlated significantly with “entrapment”, “loss of autonomy”, and “attribution: of self vs. illness” (the extent to which the individual believes that the origins of the illness lie in their personal psyche), using subscales of the Personal Beliefs about Illness Questionnaire (PBIQ). White et al. concluded that, although depression and hopelessness scores are highly correlated, the abovementioned aspects of a patient’s personal belief about their illness were all independently associated with hopelessness after controlling for depression.

Cavelti et al. [63] demonstrated that the association between insight and demoralization is mediated by the participants’ perception of their mental illness as being chronic, disabling, and out of control. The same authors [64] confirmed, in a study of 145 outpatients with schizophrenia, that the relationship between insight and demoralization was stronger as self-stigma increased, that is, patients with high insight tended to be more demoralized as a result, in part, of their increased likelihood of experiencing self-stigma in relation to their mental illness. In the same study, self-stigma also partially mediated the positive relationship between insight and demoralization, that is, the impact of insight on demoralization diminished after self-stigma was included in the regression equation. Moreover, demoralization fully mediated the adverse associations of self-stigma with psychotic symptoms and global functioning (i.e., higher levels of psychotic symptoms and lower levels of functioning were correlated with higher levels of demoralization).

Boursier et al. [65] investigated the presence of demoralization in 55 patients with schizophrenia, demonstrating that more than 94% of these patients experienced demoralization. The degree of demoralization was correlated with the intensity of positive symptoms, depression, despair, suicidality, poor quality of life, and low self-esteem.

A recent study by Wartelsteiner et al. [66] found that schizophrenia patients had significantly lower quality of life (QoL), resilience, self-esteem, and hope compared to healthy control subjects, highlighting the complex nature of QoL in patients suffering from schizophrenia and the importance of enhancing resilience and self-esteem and diminishing hopelessness and psychopathological features in patients with schizophrenia.

Recently Tourino et al. [67], in a sample of 71 outpatients with a diagnosis of schizophrenia, found that, in 21.1% of these patients, stigma was associated with suicidal ideation, a higher number of suicide attempts, higher current suicidal risk, worse self-compassion, higher scores for depression, a higher prevalence of depression, and higher levels of hopelessness.

## 4. Discussion

In this review, we have shown a clinical overlap between symptoms of demoralization and schizophrenia. Currently, it is estimated that the prevalence of depressive disorders in schizophrenia is around 40% [68,69] and the presence of depression is correlated with poorer outcomes in schizophrenia [70,71]. Bartels and Drake [27] suggested that depressive symptoms in schizophrenia included depressive symptoms secondary to organic factors, “nonorganic” depression intrinsic to the acute psychotic episode, and depressive symptoms that are not associated with the acute psychotic episode, such as symptoms associated with the prodromal, post-psychotic interval, as well as those symptoms that resemble depression that may represent negative symptoms of schizophrenia. Based on the results of this review, we suggest including the *Demoralization Syndrome* among the depressive syndromes in schizophrenia.

The demoralization syndrome can be differentiated from depression. In patients with chronic medical illness and in patients with psychiatric disorders, disability, bodily disfigurement, fear of loss of dignity, social isolation, feelings of greater dependency on others, and the perception of being a burden may predispose patients to develop symptoms of demoralization. Furthermore, patients with the demoralization syndrome, composed of the sense of impotence and helplessness, could progress to a desire to die by suicide. The research reviewed demonstrated clearly that suicide risk is associated with hopelessness and depression [21], and that suicidal intent in psychiatric inpatients correlates more strongly with hopelessness than with depression [72]. The results of this paper further support the argument for distinguishing demoralization syndrome and its constructs from clinical depression in patients affected by schizophrenia [9].

Previous research has shown that the patients with schizophrenia who are at a higher risk to die by suicide are those who are young, male, Caucasian, never married, with good premorbid function, post-psychotic depression, and with a history of substance abuse and suicide attempts. Furthermore, hopelessness, social isolation, hospitalization, deteriorating health with a high level of premorbid functioning, recent loss or rejection, limited external support, and family stress are considered important risk factors in patients with schizophrenia who die by suicide. How depression, demoralization, hopelessness, positive and negative symptoms, illness subtype, and suicidal tendency evolved in relation to each other in patients with schizophrenia is poorly understood but of clinical relevance. As shown above, several studies have highlighted the link between high levels of insight and low self-esteem and quality of life [52,53]. Patients with psychiatric illnesses experience discrimination, less life satisfaction, stigma, and feel demoralized and rejected by others [54].

The research review above shows how demoralization is frequently present in patients with schizophrenia and its significant association with suicide risk. Furthermore, hopelessness has been found to predict current and lifetime suicidality among patients with schizophrenia, independently of a diagnosis of major depression [43,44,45,46,47,48,49,50,51,52,53,54,55,56,57,58,59,60,61,62,63,64,65,66,67,68,69,70,71], emphasizing the importance of assessing demoralization symptoms in psychiatric patients to better evaluate suicide risk in clinical settings.

This review supports the hypothesis that the association between depression and suicide is moderated by hopelessness. Hopelessness may operate as a powerful potentiating variable, as demonstrated by the evidence showing that depression becomes a predictor of suicide only in the presence of hopelessness. Furthermore, the results of our review, which demonstrated the relationship between demoralization, insight, and stigma in patients affected by schizophrenia, suggest that demoralization may add valuable information about clinically relevant psychological distress in the context of psychiatric diseases. Self-insight may exert its effect on suicide risk through increasing demoralization, rather than through a direct impact on suicidal behavior. Studies in this field suggested that insight may represent a risk factor for suicide in patients with schizophrenia and this association appears to be mediated by other variables, such as depression and, above all, hopelessness. Further studies are required to analyze this issue in depth, given the crucial implications that it may have on the development of a model for suicide prevention in schizophrenia.

### Limitations

When interpreting the results of this review, several limitations should be considered. First, all reviews of research are retrospective and, therefore, are subject to bias. Second, we selected only articles in English, omitting relevant articles in other languages from our review. Third, we did not examine the role of gender on suicide risk. Fourth, it was not always simple to apply the concept of “demoralization” retroactively to studies that did not specifically set out to study it, for example ones that look at “hopelessness” or “helplessness”. Moreover, given this difficulty, the division of the two sub-chapters of the result section can sometimes seem confusing. Specifically, we included in the first section the studies that presented the main focus on suicidal risk and in the second chapter the studies that mainly considered demoralization, insight, stigma, and quality of life in patients affected by schizophrenia.

Finally, not all the studies reviewed in this paper distinguished completed suicide from attempted suicide. Suicide attempters and completers with schizophrenia appear to represent two overlapping but distinct groups, with different clinical and demographic profiles.

## 5. Conclusions

Despite increased screening for depression in clinical settings, demoralization symptoms in patients with schizophrenia are often either missed or dismissed by clinicians. This is at least in part because of the difficulty of distinguishing between symptoms of a concurrent mood disorder and those of the schizophrenia syndrome itself, in which disturbed affect and difficulty expressing emotions are central negative symptoms. Our results suggest that, in clinical practice, it is important to recognize the symptoms of demoralization, in particular hopelessness, using appropriate psychological tools in order to better approach the suffering of our patients and implement suicide prevention programs in patients with schizophrenia. Identifying specific subgroups of patients with schizophrenia with different suicide risk profiles, including the presence of demoralization symptoms, is an important goal for future research. Furthermore, in clinical practice, clinicians should definitely focus on the degree of insight in their clients in order to evaluate the suicidal risk, through a careful clinical evaluation and through the use of appropriate tools that evaluate and prevent suicidal risk.

## Figures and Tables

**Figure 1 medicina-55-00200-f001:**
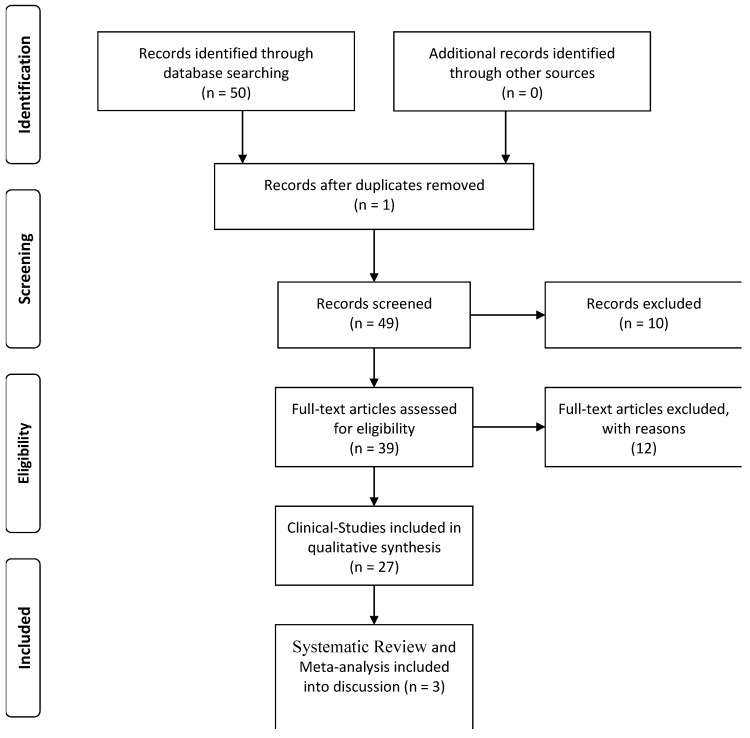
Flowchart of the search and selection process.

**Table 1 medicina-55-00200-t001:** Demoralization and Suicide Risk in Schizophrenia.

Authors	Purpose	Measures	Results/Discussion
**Drake et al. (1984)**	Risk factors for suicide in patients with schizophrenia N = 104 (15 clinical vs. 89 controls; all met DSM-III diagnostic criteria for Schizophrenia)	3d edition of the Diagnostic and Statistical Manual of Mental Disorders from the American Psychiatric Association (DSM-III)	Depressed mental status is more frequently observed in suicide group vs. non-suicide group (80% vs. 48%; *p* < 0.01).
**Cotton et al. (1985)**	Psychotherapy of suicidal schizophrenic patients. N = 20 (schizophrenia)	Semi-structured interviews	Patients with schizophrenia reported hopelessness and a strong desire to escape through death at the time of suicide. It is important to assess self-esteem, the presence of psychosis, and distinguish inability to function from unwillingness to function
**Drake & Cotton (1986)**	DSM-III depressive symptoms vs. hopelessness in schizophrenic suicides and non-suicides patients N = 104 (15 = s; 89 = n-s)	3d edition of the Diagnostic and Statistical Manual of Mental Disorders from the American Psychiatric Association (DSM-III); Beck Hopelessness Scale (BHS)	Suicide ideas, depressive mood, and hopelessness rather than depression increases risk of suicide (respectively 87%, 80%, and 67% vs. 33% of depression in suicide sample); *p* < 0.01
**Cohen et al. (1990)**	Suicide in long-term treatment schizophrenia. N = 122 (67.2% male) with Schizophrenia, schizoaffective disorder (research diagnostic criteria), or schizotypal personality disorder (DSM-III)	Symptom Checklist 90-R (SCL-90-R); Brief Psychiatric Rating Scale (BPRS); Self-report measure of satisfaction	Baseline SCL-90-R scores for: hopelessness (6.30 vs. 2.96), hostility (3.53 vs. 1.63), depression (5.89 vs. 3.52), paranoid ideation (4.51 vs. 2.71), and obsessive-compulsive (3.73 vs. 2.23) were very discriminative between suicides (N = 8) and non-suicides (N = 74)
**Fenton et al. (1997)**	Suicidality in schizophrenia and its relationship with subtypes N = 322 (schizophrenia n =187; schizoaffective disorder n = 87; schizophreniform disorder N = 15; schizotypal personality disorder n = 33)	3d edition of the Diagnostic and Statistical Manual of Mental Disorders from the American Psychiatric Association (DSM-III); Positive and Negative Symptoms Scale (PANSS); self-report interviews	Negative symptoms: blunted affect (*p* = 0.03), poverty of speech (*p* = 0.02), stereotyped thinking (*p* = 0.005), and only one of positive symptoms (grandiosity (*p* < 0.05)) were correlated with suicide behavior.
**Nordentoft et al. (2002)**	Suicidal behavior and treatment in first-episode schizophrenia (FEP) N = 341 (FEP disorder); attempts reported at baseline (N = 321); attempts during first year of treatment (N = 275) Diagnosis: schizophrenia, schizotypal disorder, delusional disorder, acute or transient psychosis, schizoaffective psychosis, induced psychosis, or unspecific non-organic psychosis according to ICD–10	Schedules for Clinical Assessment in Neuropsychiatry (SCAN 2.0), Scale for the Assessment of Negative Symptoms (SANS), Scale for the Assessment of Positive Symptoms (SAPS)	Baseline period: being male (*p* = 0.001) and suicidal plans the week prior (*p* = 0.001) were significant predictors of suicide; Follow up period: being male (*p* = 0.002), suicidal plans the week prior (*p* < 0.001), and previous suicide attempts (*p* = 0.001) were significantly associated with suicide. There was a weak association between hopelessness and suicide in the integrated treatment’s group one year after treatment compared to the standard treatment’s group (*p* < 0.01)
**Kim et al. (2002)**	Hopelessness, insight, cognitive dysfunction, and suicidality in schizophrenia N = 333 (schizophrenia)	Schedule for Affective Disorders and Schizophrenia (SADS), Hamilton Depression Rating Scale (HDRS), Brief Psychiatric Rating Scale (BPRS)	Regression analysis predicted Hopelessness as the most important predictor of current and lifetime suicidality (β = 0.41, p = 0.0001; β = 0.35, p = 0.01, respectively). Further, insight and substance abuse were predictors for lifetime and current suicidality (*p* = 0.001 and *p* = 0.004; *p* = 0.001 and *p* = 0.033, respectively).
**Tarrier et al. (2004)**	Factors (self-esteem, relatives’ expressed emotions, demographic, social, clinical) associated with suicidal ideation and/or previous suicide attempts as proxy measures of suicide risk N = 59 (DSM-IV diagnosis of recent onset schizophrenia)	4d edition of the Diagnostic and Statistical Manual of Mental Disorders from the American Psychiatric Association (DSM-IV); Modified Self-Evaluation and Social Support for Schizophrenia (SESS-sv); Positive and Negative Symptoms Scale (PANSS); Beck Depression Inventory (BDI); Beck Hopelessness Scale (BHS); Insight Scale (IS)	Greater hopelessness (OR 1.22) and longer duration of illness (OR 1.13) increase suicide risk. Hopelessness was also associated with higher negative self-evaluation and social isolation.
**Ran et al. (2005)**	Suicide attempters’ (N = 38) vs. non-attempters’ (N = 472) clinical features. N = 510 (schizophrenia)	Screening schedule for Psychosis, Present State Examination, Chinese version (PSE-9), Social Disability Screening Schedule (SDSS), General Psychiatric Interview Schedule and Summary Form	Hopelessness and depressed mood were present in 60.5% of the patients with a history of lifetime suicide attempt (*p* < 0.001).
**Montross et al. (2008)**	Prevalence and correlates of suicide in 40-year-old and older schizophrenic patients N = 132 (schizophrenia spectrum disorder and concurrent depressive symptoms)	SCID, Beck Scale for Suicidal Ideation (BSSI), Hamilton Depression Rating Scale (HAM-D), Calgary Depression Rating Scale (CDRS), Positive and Negative Symptoms Scale (PANSS), Clinical Global Impression Scale (CGI), Cumulative Illness Rating Scale-Geriatrics version C(IRS-G), Beck Hopelessness Scale (BHS)	Hopelessness rated by BHS (5.8; *p* = 0.001), and level of depression rated by HAM-D (13; *p* = 0.000) and CDR (6.4; *p* = 0.001) significantly differentiated the suicidal ideation and non-suicidal ideation groups.
**Restifo et al. (2009)**	Demoralization model (premorbid adjustment x insight) N = 164 (schizophrenia, N = 115; schizoaffective, N = 49)	Diagnostic Interview of Genetic Studies (DIGS), Premorbid Adjustment Scale (PAS)	Interaction between premorbid adjustment and insight did not significantly predict suicide attempt (*p* = 0.88, *p* =0.91)
**Pompili et al. (2009)**	Understanding suicide risk in 20 patients with schizophrenia who died by suicide vs. C 20 controls	Beck Hopelessness Scale (BHS)	Hopelessness (OR = 51.00; 95%CI = 7.56–343.72) was a risk factor for suicide
**Klonsky et al. (2012)**	Longitudinal relationship of hopelessness and attempted suicide in DSM-III-R psychotic disorders N = 414	3d edition of the Diagnostic and Statistical Manual of Mental Disorders from the American Psychiatric Association DSM-III-R; Beck Hopelessness Scale (BHS); Hamilton Depression Rating Scale (HAM-D)	Hopelessness in psychotic disorders provides information about suicide risk. In comparison to non-psychotic population, even relatively modest levels of hopelessness increase risk for suicide in psychotic patients.
**Acosta et al. (2012)**	Relationship between schizophrenic patients’ cognitions about their illness and past suicidal behaviors Relationship between patients’ beliefs about the illness with potential mediators of suicidal behaviors such as depressive symptoms, hopelessness, and insight N = 60 (ICD-10 diagnosis of schizophrenia)	International Classification of Diseases 10th Revision); (ICD-10); Calgary Depression Scale (CDS); Beck Hopelessness Scale (BHS);	Negative appraisals were associated with hopelessness and depressive symptoms (negative expectations and stigma showed the strongest associations). No differences between patients with and without past suicidal behaviors
**Fulginiti & Brekke (2015)**	Association between discrepancy factors (self-esteem and quality of life) and suicidal ideation in DSM-IV schizophrenia spectrum disorders N = 162	4d edition of the Diagnostic and Statistical Manual of Mental Disorders (DSM-IV); Extended Version of the Brief Psychiatric Rating Scale (BPRS-E); Subjective Well-being under Neuroleptic Treatment scale (SWL); Index for Self-Esteem (ISE); Medical Outcomes Study Social Support Survey (MOS-SSS); Survey); RFS (Role Functioning Scale)	QoL and self-esteem added value to predicting suicidal ideation beyond clinical and demographic factors. Self-esteem mediates the relationship between QoL and suicidal ideation.
**Yoo et al. (2015)**	Associations between suicidality and self-esteem in patients with schizophrenia according to DSM-IV N = 87 (20 attempted suicide)	4d edition of the Diagnostic and Statistical Manual of Mental Disorders (DSM-IV); Positive and Negative Symptoms Scale (PANSS); Beck Depression Inventory (BDI); Beck Hopelessness Scale (BHS); Rosenberg Self-Esteem Scale (SES); Korean version of the Internalized Stigma of Mental Illness Scale (K-ISMI)	Patients with a history of suicide attempt had significantly higher scores on BDI (*p* = 0.036) and K-ISMI (*p* = 0.009) and significantly lower scores on SES (*p* = 0.001)

**Table 2 medicina-55-00200-t002:** Demoralization, insight, and stigma in schizophrenia.

Authors	Purpose	Measures	Results/Discussion
**Birchwood et al. (1993)**	Relationship between depression and acceptance or rejection of mental illness and perceived controllability of illness in chronic psychosis N = 84 (49 schizophrenia, 35 manic-depressive disorder)	Beck Depression Inventory (BDI); Personal Beliefs about Illness Questionnaire (PBIQ); Crown Self-Esteem Scale; degree of acceptance of two statements regarding acceptance or rejection of mental illness label	Patients’ perception of controllability of their illness powerfully discriminated depressed from non-depressed psychotic patients. Label acceptance was not associated with depression, low self-esteem, or unemployment.
**Aguilar et al. (1997)**	Hopelessness in first-episode psychotic patients in 96 neuroleptic-naive psychotic patients (49 schizophrenic patients and 47 other non-affective psychotic patients)	Hopelessness Scale (HS)	High HS scores at baseline predicted poor short-term outcome in patients with schizophrenia, as evidenced by worse global functioning at the 12-month follow-up.
**Carrol et al. (2004)**	Explores the level of insight in patients with schizophrenia and its relationship to symptoms and history of violence. Relationship between the insight’s dimensions of “compliance” and “awareness of illness” and hopelessness N = 28 (DSM-IV diagnosis of schizophrenia)	4d edition of the Diagnostic and Statistical Manual of Mental Disorders (DSM-IV); Beck Hopelessness Scale (BHS); Positive and Negative Symptoms Scale (PANSS);	Awareness of illness (*p* = 0.028), but not compliance with treatment, was positively correlated with level of hopelessness.
**Lysaker et al. (2004)**	Explores two aspects of hope (expectations of the future and motivation to persist), neurocognition, personality, symptoms, and social functioning in post-acute phase of schizophrenia N = 52 (39 DSM-IV diagnosis of schizophrenia; 13 DSM-IV diagnosis of schizoaffective disorder)	4d edition of the Diagnostic and Statistical Manual of Mental Disorders (DSM-IV); Positive and Negative Symptoms Scale (PANSS); Hopkins Verbal Learning Test (HVLT); Wisconsin Card Sorting Test (WCST); (NEO Five-Factor Inventory (NEO); Vocabulary; Beck Hopelessness Scale (BHS); Quality of Life Scale (QOL)	Neuroticism, verbal memory, and income were each related to expectations of the future. Neuroticism and social isolation were related to motivational hope. Positive and negative symptoms were unrelated to either form of hopelessness.
**White et al. (2007)**	Relationship between psychiatric symptoms levels, beliefs about illness and hopelessness N = 100 (DSM-IV diagnosis of schizophrenia)	4d edition of the Diagnostic and Statistical Manual of Mental Disorders (DSM-IV); Personal Beliefs about Illness Questionnaire (PBIQ); Extended Version of the Brief Psychiatric Rating Scale (BPRS); Scale for the Assessment of Negative Symptoms (SANS); Beck Hopelessness Scale (BHS);	There were significant differences between the hopeless and non-hopeless participants on PBIQ, SANS, and BPRS. CDSS score, “humiliating need to be marginalized” PBIQ subscale and BPRS score contributed significantly (60% of the variance) to hopelessness scores.
**Cavelti et al. (2012)**	Mechanisms underlying the association of insight, depressive symptoms, and protective factors in patients with DSM-IV diagnosis of schizophrenic spectrum disorders N = 142	4d edition of the Diagnostic and Statistical Manual of Mental Disorders (DSM-IV); Scale to Assess Unawareness of Mental Disorder (SUMD); Beck Depression Inventory (BDI-II); Subjective Well-being under Neuroleptic Treatment scale (SWN); Illness Perception Questionnaire for Schizophrenia (IPQS);	Higher levels of insight and psychotic symptoms were associated with more depressive symptoms. Participants’ perception of their illness as chronic and disabling mediates the relationship between insight and depressive symptoms. The association of insight and depressive symptoms was less pronounced in patients with positive recovery attitude
**Cavelti et al. (2012)**	Investigates self-stigma both as a moderator and as a mediator variable in the relationship between insight and demoralization in patients with DSM-IV diagnosis of schizophrenic spectrum disorder N = 145	4d edition of the Diagnostic and Statistical Manual of Mental Disorders (DSM-IV); Scale to Assess Unawareness of Mental Disorder (SUMD); Beck Depression Inventory (BDI); Subjective Well-being under Neuroleptic Treatment scale (SWN); Self-Stigma of Mental Illness Scale (SSMIS);	The association of insight and demoralization was stronger as self-stigma increased, confirming self-stigma as a moderator. Self-stigma also partially mediates the positive relationship between insight and demoralization. Demoralization fully mediates the adverse associations of self-stigma with psychotic symptoms and global functioning
**Boursier et al. (2013)**	Demoralization in psychotic patients N = 55 (schizophrenic disorder)	Positive and Negative Symptoms Scale (PANSS), Beck Hopelessness Scale (BHS), Poor quality of life (SqOL)	94% of the sample was found to be demoralized. Demoralization correlates with positive symptoms (*p* = 0.016), depression (*p* < 0.001), despair (*p* = 0.015), suicidality (*p* < 0.01), and poor quality of life (*p* = 0.007).
**Vass et al. (2015)**	Impact of stigma on symptomatic and subjective recovery from psychosis (currently and longitudinally). Investigates whether self-esteem and hopelessness mediate the association between stigma and outcomes N = 80	International Classification of Diseases 10th Revision (ICD-10); Beck Hopelessness Scale (BHS); Questionnaire about the Process of Recovery (QPR); Positive and Negative Symptoms Scale (PANSS)	Stigma predicted both symptomatic and subjective recovery. Hopelessness and self-esteem mediated the effect of stigma on symptomatic and subjective recovery. At the follow-up, stigma predicted recovery and symptoms.
**Wartelsteiner et al. (2016)**	Examines the correlation of resilience, self-esteem, hopelessness, and psychopathology with quality of life N = 129 (52 DSM-IV diagnosis of schizophrenia; 77 healthy controls)	4d edition of the Diagnostic and Statistical Manual of Mental Disorders	Patients with schizophrenia had lower levels of QoL, resilience, self-esteem, and hope than healthy control subjects. In these patients, QoL correlated moderately with resilience, self-esteem, and hopelessness and weakly with symptoms (negative correlation with depression and positive symptoms)
**Touriño et al. (2018)**	Assesses prevalence of internalized stigma in patients with ICD-10 diagnosis of schizophrenia who attend psychosocial rehabilitation programs. Investigate the relationship between internalized stigma and sociodemographic, general clinical, psychopathologic, psychological, and suicidal behaviour variables in schizophrenic patients N = 71	ICD-10; ISMI; RSES; SUMD; BHS; CGI-SCH; CDS; SCS	Stigma was associated with higher prevalence of suicidal ideation, a higher number of suicide attempts, higher current suicidal risk, worse self-compassion, higher self-esteem, higher scores on depression, higher prevalence of depression, and higher hopelessness. Hopelessness and the existence of depression were independently associated with internalized stigma

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
