# Peer review of "The Role of Demoralization and Hopelessness in Suicide Risk in Schizophrenia: A Review of the Literature"

_medicina, 2019, doi:10.3390/medicina55050200_

Round 1

Reviewer 1 Report

In this manuscript, Berardelli et al present a systematic review, produced using PRISMA guidelines, into clinic reports of demoralization and hopelessness in schizophrenia patients – with a specific aim of investigating their relationship to the risk of suicide.

The literature review itself appears to have been carried out in a logical and systematic manner, with all articles being reviewed by multiple authors in order to increase consistency. There is no evidence of obvious flaws in their review process, or of any bias. The process can be assumed to be reliable. The 27 studies included in the final review are discussed in the text and presented in tabulated form. The final discussion is fair and conclusions reasonable. The authors present and briefly discuss the limitations of their study. I therefore have no major concerns regarding the review.

Minor concerns:

1) The division of the papers into two sections/tables (one on demoralization and suicide risk, the other on demoralization, insight and stigma) is useful in synthesizing the results of the papers, although the distribution of papers between the two can be a little arbitrary. For example, why were Ran et al (2005) or Boursier et al (2013) put in table 2 rather than table 1?

2) The author affiliation list appear to contain a misspelling of “Psychology”

3) In figure 1, “n” values of papers are given at each stage of the search and selection process. However in some cases, the number remaining is listed, and sometimes the number remaining is listed – making it somewhat confusing to read. Perhaps only papers remaining should be listed in the main column?

4) Perhaps the limitations section should also contain a couple of sentences about how difficult or simple it is to apply the concept of “demoralization” retroactively to studies in which did not specifically set out to study it (for example ones that look at “hopelessness”).

Author Response

Response to Reviewer 1 Comments

Minor concerns:

Point 1) The division of the papers into two sections/tables (one on demoralization and suicide risk, the other on demoralization, insight and stigma) is useful in synthesizing the results of the papers, although the distribution of papers between the two can be a little arbitrary. For example, why were Ran et al (2005) or Boursier et al (2013) put in table 2 rather than table 1?

Response 1: Thank you very much for the advice, according to your suggestion we moved Ran et al. 2005 in the first section of the paper and then in the first table, while Boursier et al. 2013 we left it in the second section because the work emphasizes the altered quality of life rather than suicidal risk. Furthermore, we explained this point in the “limitations of the study”.

Point 2) The author affiliation list appear to contain a misspelling of “Psychology”

 Response 2: We have now edit the term “Psychology”

Point 3) In figure 1, “n” values of papers are given at each stage of the search and selection process. However in some cases, the number remaining is listed, and sometimes the number remaining is listed – making it somewhat confusing to read. Perhaps only papers remaining should be listed in the main column?

Response 3: Thank you very much for the comment, for figure 1 we followed the guidelines for Preferred Reporting Items for Systematic Reviews and Meta-Analyses (PRISMA), in particular the Flow            Diagram From: Moher D, Liberati A, Tetzlaff J, Altman DG, The PRISMA Group (2009). Preferred Reporting Items for Systematic Reviews and MetaAnalyses: The PRISMA Statement. PLoS Med 6(7): e1000097. doi:10.1371/journal.pmed1000097. Furthermore we now modified the figure 1 introducing the number of clinical studies included in the review and we introduced in the method section the following sentence for better explain the flow chart figure: “In the result session we excluded cases report, meta-analyses and systematic reviews, and studies which did not clearly report statistical analysis, diagnostic criteria, or the number of patients included. We found 27 articles on demoralization and schizophrenia that met these criteria and we discussed them in the two sessions of the results (Clinical Studies included in qualitative synthesis.) We have included 3 systematic reviews exclusively in the discussion section but not in the results section”.

Point 4) Perhaps the limitations section should also contain a couple of sentences about how difficult or simple it is to apply the concept of “demoralization” retroactively to studies in which did not specifically set out to study it (for example ones that look at “hopelessness”).

Response 4: In accordance with your advice we have introduced a sentence in the limitation section on the difficulty to apply the concept of “demoralization” retroactively to studies in which did not specifically set out to study it (for example ones that look at “hopelessness”).

Reviewer 2 Report

Thank you for the opportunity to review “The Role of Demoralization and Hopelessness in Suicide Risk in Schizophrenia: A Review of the Literature.”  I commend the authors for examining an important topic with great clinical relevance.  The following feedback is intended to assist the authors in refining their manuscript.

The authors do a nice job of defining demoralization in their Introduction and cite various ways it has been conceptualized by different authors.  

The first line in the Methods section indicates that demoralization was defined so as to include inability to cope, helplessness, hopelessness, and low self-esteem.  Many of the studies included in the review assess individual variables that are associated with, and part of, demoralization (e.g., hopelessness, depressed mood), but it’s not clear that they assess demoralization as a constellation of the 4 variables listed in the authors’ definition.  Demoralization is a complicated construct – a syndrome, as the authors argue – but the articles reviewed seem to be examining pieces of that syndrome.  The authors do mention that they examined studies that have investigated “the presence of demoralization (and its constructs)” in line 114.  This phrase acknowledges that the studies are not necessarily examining demoralization in its totality, but I would suggest that the authors discuss this in their limitations section – e.g., state that their review examined studies that included variables relevant to demoralization, but each study did not examine the construct of demoralization as a whole. 

Figure 1 could be edited for clarity.  It is not clear what the authors mean by “Clinical Studies included in qualitative synthesis.”  Do they mean the 27 studies they chose to include in the review?  Also, what are the authors referring to when they write “Systematic Review and meta-analysis included in discussion (n = 3)?” How does this figure into the 27 studies included in this article?  The authors state on p. 3, line 132 that meta-analyses and systematic reviews were excluded from review.  Please explain.

In Tables 1 and 2, for the second column entitled “Outcomes and Measures,” the authors should consider including the entire names of the measures (with abbreviations following in parentheses) rather than listing the names right after the table.  Also, the column should be titled “Measures” rather than “Outcomes and Measures” since no outcomes are discussed in this column.

While I commend the authors for attempting to organize the results with subheadings (3.1 and 3.2), the division of the studies into two sections is not completely clear.  For example, why not include the Aguilar and Ran studies (p. 8) under Section 3.1 “Demoralization and suicide risk in schizophrenia”?  It’s not clear that these studies include variables relevant to insight and stigma.  Why are they included under the heading 3.2 “Demoralization, insight, stigma in schizophrenia”?   

In the discussion section, it would be helpful to discuss clinical implications regarding the role of insight, since you found a link between greater client insight and higher suicide risk.  What knowledge should a clinician take away from your review regarding client insight?  Therapists will often try to enhance a client’s insight; however, your review suggests that a more realistic sense of one’s difficulties is associated with elevated suicide risk.  What can a clinician do with this information?

Author Response

Response to Reviewer 2 Comments

Point 1: The first line in the Methods section indicates that demoralization was defined so as to include inability to cope, helplessness, hopelessness, and low self-esteem.  Many of the studies included in the review assess individual variables that are associated with, and part of, demoralization (e.g., hopelessness, depressed mood), but it’s not clear that they assess demoralization as a constellation of the 4 variables listed in the authors’ definition.  Demoralization is a complicated construct – a syndrome, as the authors argue – but the articles reviewed seem to be examining pieces of that syndrome.  The authors do mention that they examined studies that have investigated “the presence of demoralization (and its constructs)” in line 114.  This phrase acknowledges that the studies are not necessarily examining demoralization in its totality, but I would suggest that the authors discuss this in their limitations section – e.g., state that their review examined studies that included variables relevant to demoralization, but each study did not examine the construct of demoralization as a whole. 

Response 1: Thank you very much for your suggestion. We have introduced the following sentence in the limits section of the study:  Fourth, it was not always simple to apply the concept of “demoralization” retroactively to studies in which did not specifically set out to study it, for example ones that look at “hopelessness” or “helplessness”. Moreover, given this difficulty, the division of the two sub-chapters of the result section can sometimes seem confusing. Specifically, we included in the first section the studies that presented the main focus on suicidal risk and in the second chapter the studies that mainly considered demoralization, insight, stigma and quality of life in patients affected by schizophrenia.

Figure 1 could be edited for clarity.  It is not clear what the authors mean by “Clinical Studies included in qualitative synthesis.”  Do they mean the 27 studies they chose to include in the review?  Also, what are the authors referring to when they write “Systematic Review and meta-analysis included in discussion (n = 3)?” How does this figure into the 27 studies included in this article?  The authors state on p. 3, line 132 that meta-analyses and systematic reviews were excluded from review.  Please explain.

Response 2: According with your suggestion, we edit the method section for better explain the flow chart figure: In the result session we excluded cases report, meta-analyses and systematic reviews, and studies which did not clearly report statistical analysis, diagnostic criteria, or the number of patients included. We found 27 articles on demoralization and schizophrenia that met these criteria and we discussed them in the two sessions of the results. We have included 3 systematic reviews exclusively in the discussion section but not in the results section. The principal reviewer (IB) checked all items.

In Tables 1 and 2, for the second column entitled “Outcomes and Measures,” the authors should consider including the entire names of the measures (with abbreviations following in parentheses) rather than listing the names right after the table.  Also, the column should be titled “Measures” rather than “Outcomes and Measures” since no outcomes are discussed in this column.

Response 3: Thank you for your suggestion, we now edit the column “Outcomes and Measures” in table 1 and table 2 and we included the entire names of measures in the two tables.

While I commend the authors for attempting to organize the results with subheadings (3.1 and 3.2), the division of the studies into two sections is not completely clear.  For example, why not include the Aguilar and Ran studies (p. 8) under Section 3.1 “Demoralization and suicide risk in schizophrenia”?  It’s not clear that these studies include variables relevant to insight and stigma.  Why are they included under the heading 3.2 “Demoralization, insight, stigma in schizophrenia”?   

Response 4: Thank you for your comments. We now organized the results with subheadings (3.1, 3.2). We agree that the division of the studies into two sections is not completely clear and we now discussed this point in the limitation sections. We now change Ran et al. (p.8) in the first section and in the first table, however we the study by Aguilar since it does not take suicide risk into consideration, we have not changed it.

In the discussion section, it would be helpful to discuss clinical implications regarding the role of insight, since you found a link between greater client insight and higher suicide risk.  What knowledge should a clinician take away from your review regarding client insight?  Therapists will often try to enhance a client’s insight; however, your review suggests that a more realistic sense of one’s difficulties is associated with elevated suicide risk.  What can a clinician do with this information?

Response 5: According with your comments we now introduce the following sentences in the discussion: Studies in this field suggested that insight may represent a risk factor for suicide in patients with schizophrenia and this association appears to be mediated by other variables such as depression and, above all, hopelessness. Further studies are required to analyze this issue in depth, given the crucial implications that it may have on the development of a model for suicide prevention in schizophrenia.
